# Genomic versus Plasmid-Borne Expression of Germinant Receptor Proteins in *Bacillus* *cereus* Strain 14579

**DOI:** 10.3390/microorganisms10091774

**Published:** 2022-09-02

**Authors:** Yan Wang, Peter Setlow, Stanley Brul

**Affiliations:** 1Molecular Biology and Microbial Food Safety, Swammerdam Institute for Life Sciences, University of Amsterdam, Science Park 904, 1098 XH Amsterdam, The Netherlands; 2Department of Molecular Biology and Biophysics, UConn Health, Farmington, CT 06030-3305, USA

**Keywords:** *Bacillus* *cereus*, spore, germinant receptor, genome integration, inducible high(over) expression

## Abstract

Germinant receptors (GRs) are proteins in the spore-forming bacteria of Bacillus species that are crucial in triggering spore germination by sensing nutrients in the spores’ environment. In the Gram-positive bacterium *Bacillus cereus* strain ATCC 14579, the GerR GR initiates germination with L-alanine. While we have expressed GerR subunits fused to reporter proteins from genes under control of their native promoter on plasmids in this *B. cereus* strain, here we sought increased flexibility in this work by studying genome integration and plasmid-borne inducible high level (over) expression. However, construction of chromosomal integrants to visualize and localize the GerR B subunit fused to fluorescent reporter protein SGFP2 was not successful in this *B. cereus* strain using constructs with either shorter (~600 bp) or longer (~1200 bp) regions of homology to the *gerR* operon. This failure was in contrast to successful IPTG-inducible expression of GerRB-SGFP2 from plasmid pDG148 in vegetative cells and dormant spores, as fluorescent GerRB-SGFP2 foci were present in vegetative cells and the protein was detected by Western blot analysis. In dormant spores, the fluorescence intensity with IPTG-inducible expression from pDG148-*gerRB*-*SGFP2* was significantly higher than in wild type spores. However, the full length GerRB-SGFP2 protein was not detected in spores using Western blots. Clearly, there are still challenges in the construction of *B. cereus* strains harboring fluorescent reporter proteins in which tagged proteins are encoded by genes incorporated in the chromosome or on extrachromosomal expression plasmids.

## 1. Introduction

*Bacillus cereus*, a food poisoning bacterium in humans, was first isolated in 1887 [1] and viewed as a pathogen in 1963 [2]. This bacterium can be isolated from many sources or environments including dairy food products, soil-dwelling arthropods or insects, the plant rhizosphere, and tidal flat sediments [3,4,5]. This Gram-positive endospore former is a member of the *B. cereus* sensu lato group, which contains *Bacillus thuringiensis*, *Bacillus anthracis*, *Bacillus mycoides*, *Bacillus weihenstephanensis*, *Bacillus pseudomycoides*, *Bacillus toyonensis*, and *Bacillus cytotoxicus*, as well as more recently identified members, such as *Bacillus wiedmannii*, *Bacillus manliponensis*, and *Bacillus gaemokensis* [3,5,6].

Spores of Bacillus can survive for a long time in the dormant state and even in harsh environmental conditions [7]. However, dormant spores can sense low molecular weight nutrients, such as amino acids and sugars, by germinant receptors (GRs) localized in the spores’ inner membrane (IM), and then initiate spore germination [8]. Initiation of germination of *B. cereus* spores can be triggered by L-alanine, L-threonine, L-glutamine, or inosine. The spores of *B. cereus* strain 14579 have seven *ger* operons that encode GRs: GerR, GerK, GerG, GerL, GerQ, GerI, and GerS [9]. GerG responds to L-glutamine, GerI and GerQ respond to inosine [10], and GerR, encoded by the *gerR* operon, responds to L-alanine and has an important role in the germination of these spores [11]. In general, each GR contains A, B, and C subunits. The A and B subunits have large integral membrane regions, but the C subunit is a lipoprotein anchored in spores’ IM with an N-terminal lipid moiety [12]. While we have successfully expressed GR-reporter proteins from genes under control of their native promoter on plasmids in *B. cereus* 14579 [13,14], here we sought to increase flexibility in this work by studying genome integration on the one hand and plasmid-borne inducible high (over)expression on the other hand.

Previous work shows that *B. cereus* 14579 has a natural competence that allows it to take up foreign DNA and integrate it into its genome using the transcriptional factor ComK [15,16]. This property suggests that this *B. cereus* strain could be used to knock-out or knock-in specific genes or even inducible overexpression systems. For instance, the *gerR* operon was proven to be involved in germination triggered by L-alanine using the integration vector pMUTIN4 [11]. There are also many fluorescent proteins suitable for the generation of fusion proteins in *B. cereus*, for example, the integration vector pSG1151 used to perform a single crossover event in the region upstream of the *cotD* gene driven by the expression of *gfpmut1* [17]. A third integration vector pSG1164 contains an inducible xylose promoter and has been used to carry out the integration of genomic DNA into the model spore-former, *Bacillus subtilis* [18], and the shuttle vector pDG148 contains a hybrid promoter P*spac*, which can be replicated and selected for in *E. coli* and *B. subtilis* [19].

Fusions to the strongly enhanced green fluorescent protein, SGFP2, which is more stable and brighter than wild type GFP, were used to localize GR proteins in dormant spores of *B. cereus* 14579 in previous work [20]. In the current work, we aimed to visualize and express the GerR B subunit under IPTG-inducible control in vegetative cells and dormant spores of *B. cereus* 14579. The integration vectors pSG1151 and pSG1164 were also used in attempts to obtain single crossover events when carrying 0.6 or 1.2 kb homologous regions to the *gerR* region of the *B. cereus* genome, with DNA introduced into the competent cells via electroporation. The expression of a GerRB-SGFP2 fusion protein was also induced by IPTG from plasmid pDG148 in *B. cereus* vegetative cells and dormant spores, visualized by fluorescence microscopy, and protein expression detected by Western blot analysis. The results with smaller and larger homologous regions showed that integration into the genome did not occur using either vectors pSG1151 or pSG1164. The latter negative results suggest that it is difficult to carry out integration into the genome of *B. cereus* 14579 and that more work is needed to learn how to readily promote such integrations. However, the GerRB-SGFP2 fusion protein was expressed by induction in *B. cereus* 14579 carrying plasmid pDG148-*gerRB*-*SGFP2* and was present as fluorescent foci that were readily seen in vegetative cells, although not in dormant spores. The full length GerRB-SGFP2 fusion protein was also detected in vegetative cells, but not in dormant spores. These results suggest *that gerRB-SGFP2* can be induced via IPTG-induced gene-expression, but because of the high amount of fusion protein and/or the instability of GerRB in developing spores without equal amounts of GerRA and GerRC subunits, the full-length fusion protein is not assembled and localized to the IM of *B. cereus* 14579 spores.

## 2. Materials and Methods

### 2.1. Plasmid Construction for Chromosomal Integration

The procedure for isolation of *B. cereus* 14579 genomic DNA was performed as in previous work [13]. The reference sequence of this *B. cereus* genomic DNA in GeneBank is AE016877. The integration vectors used in this study are listed in Table 1, and primers used to construct the integration plasmids are listed in Table 2. A 678 bp fragment of the *gerR* operon was amplified from genomic DNA with primers YW_1 and YW_2 and the amplicon named 0.6 *gerR* (Figure 1). The 1234 bp fragment of the *gerR* operon was amplified from genomic DNA with primers YW_5 and YW_2 and the amplicon named 1.2 *gerR* (Figure 1). The *SGFP2* gene was amplified from plasmid pSGFP2-C1 with primers YW_3 and YW_4 [20]. The purified 0.6 *gerR* and *SGFP2* amplicons were fused with a flexible linker (GAG) using a two-step fusion PCR. The purified 0.6 *gerR*-*SGFP2* fusion fragment was cloned into vectors pSG1151 and pSG1164 between the *Kpn* I and *Spe* I sites, resulting in plasmids pSG1151-0.6 *gerR*-*SGFP2* and pSG1164-0.6 *gerR*-*SGFP2*, respectively (Figure 2A). The purified 1.2 *gerR* and *SGFP2* were fused with a flexible linker (GAG) using a two-step fusion PCR, and the purified fusion fragment was cloned into vectors pSG1151 and pSG1164 between the *Kpn* I and *Spe* I sites. This ligation product was transformed into competent *E. coli* DH5α cells [21], and selection of positive clones resulted in plasmids pSG1151-1.2 *gerR*-*SGFP2* and pSG1164-1.2 *gerR*-*SGFP2*, respectively (Figure 2A). The correct construction of all recombinant plasmids was confirmed using double restriction enzyme digestion and DNA sequencing (Macrogen Europe B. V., Amsterdam, The Netherlands).

### 2.2. Construction of Plasmids for IPTG-Inducible Gene Expression

The expression vector pDG148 that can replicate in *B. cereus* and contains the IPTG-inducible P*spac* promoter was used to construct plasmids for inducible gene expression. The plasmids derived from pDG148 are listed in Table 1, and primers used are listed in Table 2. The 1107 bp fragment with the complete *gerRB gene* (BC_0782) was amplified from genomic DNA with primers YW_6 and YW_2 and the amplicon named *gerRB*. The *SGFP2* gene was amplified from plasmid pSGFP2-C1 with primers YW_3 and YW_7. The purified *gerRB* and *SGFP2* fragments were fused with a flexible linker using a two-step fusion PCR. The purified *gerRB*-*SGFP2* fragment was cloned into pDG148 between the *Sal* I and *Sph* I sites, producing pDG148-*gerRB*-*SGFP2* (Figure 2B). The correct construction of the recombinant plasmids was confirmed using analysis of double restriction enzyme digestion and DNA sequencing (Macrogen Europe B. V. The Netherlands).

### 2.3. Electroporation and Growth of B. cereus 14579

Electroporation of *B. cereus* 14579 competent cells was conducted as described previously [13]. Chloramphenicol- or kanamycin-resistant single colonies from transformations with the vectors with backbones of pSG1151, pSG1164, and pDG148 were analyzed by colony PCR with the sequencing primers YW_8/YW_9, YW_10/YW_11, and YW_12/YW_13, respectively. *B. cereus* was grown in trypticase soy broth (TSB) medium at 30 °C and, when needed, 4 µg/mL chloramphenicol or 25 µg/mL kanamycin were present.

### 2.4. Preparation of IPTG-Induced Vegetative Cells and Dormant Spores

Strains of *B. cereus,* wild type and containing plasmid pDG148-*gerRB*-*SGFP2,* were grown in TSB medium and TSB medium with 25 µg/mL kanamycin, respectively, at 30 °C at 200 rpm overnight. The following day, the overnight cultures were diluted 1/100 in TSB medium without (wild type) or with (plasmid-containing strain) kanamycin, and growth continued until the OD600 reached 0.4. The culture was then made 4 mM in IPTG (isopropyl-β-D-thiogalactopyranoside) and grown for 24 h at 30 °C and 200 rpm to induce the expression of the GerRB-SGFP2 fusion protein, and growing cells were harvested for protein extraction.

Sporulation of *B. cereus* strains used in this study was performed in a chemically defined growth and sporulation (CDGS) medium, which contained the following components (final concentrations): D-glucose (10 mM), L-glutamic acid (20 mM), L-leucine (6 mM), L-valine (2.6 mM), L-threonine (1.4 mM), L-methionine (470 μM), L-histidine (320 μM), DL-lactate sodium (5 mM), acetic acid (1 mM), FeCl_3_·6H_2_O (50 μM), CuCl_2_·2H_2_O (2.5 μM), ZnCl_2_ (12.5 μM), MnSO_4_·H_2_O (66 μM), MgCl_2_·6H_2_O (1 mM), (NH_4_)_2_SO_4_ (5 mM), Na_2_MoO_4_·2H_2_O (2.5 μM), CoCl_2_·6H_2_O (2.5 μM), and Ca(NO_3_)_2_·4H_2_O (1 mM); harvesting and purification of spores was completed as described in previous work [13]. Note that 4 mM IPTG was used in CDGS medium to induce the expression of GerRB-SGFP2 fusion protein during sporulation. The purified dormant spores were stored at 4 °C prior to analysis of spores and spore proteins.

### 2.5. Extraction of Total Protein and Western Blot Analysis

Extracts of vegetative cells and dormant spores were prepared by methods described previously [13,22,23]. Briefly, vegetative cells or dormant spores (OD600 = 25) were suspended in 200 µL TEP buffer (50 mM Tris-HCl (pH7.4), 5 mM dipotassium-EDTA, cOmplete™ EDTA-free Protease Inhibitor Cocktail (Sigma Aldrich Chemie B. V. The Netherlands, Cat. No.: 4693132001), 1% Triton X−100) containing 0.5 mg/mL lysozyme (Sigma Aldrich Chemie B. V. The Netherlands, Cat. No.: L6876). The extract was generated by bead beating with 800 mg of 0.1 mm diameter Zirconia/Silica beads (BioSpec Products, Bartlesville, OK, USA) at 6000 rpm for 30 sec with six rounds and cooling on ice between each round, in a Precellys 24 tissue homogenizer (Bertin Instruments, Montigny-le-Bretonneux, France). The extract was incubated with 1% sodium dodecyl sulfate (SDS) and 150 mM 2-mercaptoethanol (Bio-Rad Laboratories B.V., The Netherlands. Cat No.:1610710) for 2 h at room temperature. The sample was bead-beaten twice more for 30 sec each at 6000 rpm to further disrupt aggregates, and 150 μL more TEP buffer was added to further suspend the sample. Finally, the sample was centrifuged at 21,500× *g* and 4 °C for 15 min, and the supernatant fluid was stored. The concentration of total protein in the extract was determined by the Pierce™ BCA Protein Assay Kit (Thermofisher Scientific, The Netherlands, Cat. No.: 23227). Western blot analysis was performed using a 10% MiniPROTEAN^®^ TGX™ gel (Bio-Rad, Laboratories B. V., The Netherlands, Cat. No.: 4561034), a 0.45 μm PVDF membrane, a primary antibody of rabbit polyclonal anti-GFP antibody at a 1:2500 dilution (abcam, ab290), and a secondary antibody of HRP-conjugated goat anti-rabbit IgG H&L at a 1:5000 dilution (abcam, ab205718), as described previously [13].

### 2.6. Imaging Settings and Analysis

Preparation of microscope slides and imaging were carried out as described previously [24]. Imaging of vegetative cells and dormant spores was performed with a Nikon Eclipse Ti microscope equipped with phase-contrast and fluorescence components. For phase-contrast imaging, a CFI Plan Apochromat Lambda 100× Oil, ORCA-Flash 4.0 Digital CMOS camera C11440–22CU (Hamamatsu Photonics K.K, Sewickley, PA, USA) with a Prior Brightfield LED and Lambda 10-B shutter (Sutter Instruments, Novato, CA, USA) was equipped with NIS-elements 4.50 version. The fluorescence visualization of vegetative cells and dormant spores of *B. cereus* was carried out with excitation at 470 ± 12 nm and emission at 516 ± 24 nm. All microscope images were analyzed with Fiji/ImageJ software (http://rsbweb.nih.gov/ij, accessed on 10 September 2021), and the spores’ fluorescence intensity was measured by the plugin SporeAnalyzer_1c.ojj (https://sils.fnwi.uva.nl/bcb/objectj/examples/SporeAnalyzer/MD/SporeAnalyzer.html, accessed on 10 September 2021). All statistical analyses were carried out using GraphPad 9.0 version.

## 3. Results

### 3.1. Confirmation of Recombinant Plasmids and the pDG148-Derived B. cereus GerRB-SGFP2 Strain

In this study, we aimed to perform a single crossover at the *gerR* operon in the *B. cereus* chromosome using integration vectors pSG1151 and pSG1164 containing *gerR* DNA (Figure 2A; Table 1). The correct construction of recombinant plasmids was confirmed using DNA sequencing and digestion by *Kpn* I and *Spe* I. The latter treatment gave the expected band of ~1.4 kb from pSG1151-0.6 *gerR*-*SGFP2* and pSG1164-0.6 *gerR*-*SGFP2* (Figure 3A,B), and at 2.0 kb from pSG1164-1.2 *gerR*-*SGFP2* (Figure 3C). Unfortunately, no clones with chloramphenicol resistance were obtained after electroporation of competent cells of *B. cereus* 14579 with plasmid pSG1164-1.2 *gerR*-*SGFP2* (Figure 3D); similar negative results were obtained with plasmids pSG1151-0.6 *gerR*-*SGFP2*, pSG1164-0.6 *gerR*-*SGFP2*, and pSG1151-1.2 *gerR*-*SGFP2* (data not shown). These results indicate that integration by a single crossover in the *gerR* region of the genome of *B. cereus* 14579 was not suitable to knock-in a foreign reporter gene.

In contrast to the negative results described above, *Kpn* I and *Spe* I digestion of plasmid pDG148-*gerRB*-*SGFP2* gave the expected *gerRB*-*SGFP2* fusion fragment of ~1.8 kb (Figure 3E), and colony PCR of *B. cereus* transformed with this plasmid showed the expected 1.8 kb fragment in 1% gel electrophoresis (Figure 3F). These results indicate that the *B. cereus* strain carrying plasmid pDG148 with *gerRB*-*SGFP2* was successfully constructed and ready to use for GerRB visualization and detection.

### 3.2. Visualization and Detection of the GerRB-SGFP2 Fusion Protein in Vegetative B. cereus Cells

*B. cereus* pDG148-GerRB-SGFP2 was constructed to express the GerRB-SGFP2 fusion protein from the IPTG-inducible P*spac* promoter. Notably, vegetative cells without the plasmid exhibited no fluorescent foci when compared to cells with the plasmid (Figure 4A). Note that previous work [13] also showed that an empty plasmid had no effect on the fluorescence of *B. cereus* spores compared to that of wild type spores. However, growth of the plasmid-containing strain carrying *gerRB-SGFP2* with IPTG gave more and brighter fluorescent foci than growth without IPTG (Figure 4Ad,Af). To examine the expression of GerRB-SGFP2, Western blot analysis was carried out and showed that the expected ~70 kDa band was detected in vegetative *B. cereus* cells with pDG148-GerRB-SGFP2, and with or without IPTG added (Figure 4B). Such leaky expression from the P*spac* promoter has been seen previously in *E. coli* and *B. subtilis* [25,26]. A nonspecific ~55 kDa protein also cross-reacted with the polyclonal antibodies used and bands of ~37 and 17 kDa also appeared to contain GFP antigenic material, and have been presumably generated by digestion of the 70 kDa intact fusion protein in extracts.

### 3.3. Analysis of GerRB-SGFP2 Expression in Dormant B. cereus Spores

The GerR GR of *B. cereus* was previously localized in the spore IM [13]. In the current study, we aimed to visualize and detect IPTG-induced GerRB-SGFP2 expression in spores using phase contrast microscopy and Western blot analysis, and using *B. cereus* pDG148-*gerRB*-*SGFP2* dormant spores prepared with and without IPTG. The results show that the SGFP2 fluorescence was spread throughout the spore when spores were prepared with IPTG, and no bright, clear foci were detected, in contrast to previous work (Figure 5A) [13]. The fluorescence intensity of spores prepared with 4 mM IPTG was significantly higher than in wild type spores, as well as in plasmid-carrying spores prepared without IPTG (0 mM). However, the expected ~70 kDa GerRB-SGFP2 band was not detected in Western blot analysis of proteins from spores prepared with or without IPTG (Figure 5C), although a possible band of the ~27 kDa SGFP2 was detected in extracts from spores prepared with or without IPTG (Figure 5C). A nonspecific 100 kDa protein in extracts also cross-reacted with the polyclonal antibodies used, as did proteins of ~30 and ~55 kDa (Figure 5C).

## 4. Discussion

In previous studies, several researchers were successful in constructing genetically altered *B. cereus* strains, including 14579, using: (i) the shuttle vector pAD123 of *E*. *coli* or Gram-positive bacteria containing reporter gene *gfpmut3a* to study the promoter activity of target genes in vegetative cells [17], (ii) the integration vector pSG1151 to knock-in a reporter gene into a target homology region in a coat gene of a spore [17], and (iii) the low copy number pHT315 plasmid to visualize the GerP protein in spores’ inner coat [27]. However, most studies focused on genes expressed in vegetative cells or on spore coat proteins. In this work, we tried to construct strains to knock-in a fluorescent reporter gene into the 3′-end of the *gerR* operon in spores of *B. cereus* 14579, but a single crossover event to do this was unsuccessful using several different-sized homology regions. Thus, it seems that integration vectors suitable for this type of procedure in *B. subtilis* are not effective in *B. cereus*, at least for strain 14579. There has been success in constructing several types of mutants in the *B. cereus* strain ATCC 10876 by transformation with recombinant transposons or plasmids [28,29,30,31]. However, these latter mutants focused only on disrupting or knocking-out target genes using transposon Tn*917* and the integration vector pMUTIN4. We have searched the literature to try and find published successes in inserting a reporter protein into a target gene in *B. cereus*, but found nothing. This failure then suggests that more work is needed to obtain an effective universal integration vector in *B. cereus*.

Previously, the shuttle vector pDG148 has been used in *B. subtilis*, the model Gram-positive spore former, to induce the overexpression of: (i) RodZ and PgsA genes involved in membrane homeostasis [32]; (ii) YabA, a negative regulator of the initiation of DNA replication [33]; and (iii) an improved pH-sensitive fluorescent protein, pHluorin, to monitor the pH in living cells [34]. In this work, a strain of *B. cereus* expressing the GerRB-SGFP2 fusion protein was constructed using the *E. coli*/*B. subtilis* expression vector pDG148 containing a P*spac* promoter. Our data show that the GerRB-SGFP2 fusion protein can be visualized and detected in vegetative cells of *B. cereus* with pDG148-*gerRB*-*SGFP2* grown with IPTG and thus the P*spac* promoter works not only in *B. subtilis* but also in *B. cereus*, although there was leaky expression in the absence of IPTG. This leaky expression has also been seen in *B. subtilis*, and in order to solve this problem, the pDG148-Stu plasmid, a modified pDG148, was created [35]. It is possible that the use of the pDG148-Stu vector might allow IPTG-inducible GerRB-SGFP2 expression without expression in the absence of IPTG in future work.

In previous work, we found that GerRB stability in developing spores was improved when the B subunit was co-expressed with the A and C subunits [13]. In the current work we found that GerRB-SGFP2 was expressed from a plasmid in growing cells, and a full-length fusion protein of the expected size was detected by Western blot analysis of the total amount of vegetative cell protein, although there was also smaller anti-GFP reactive material. Notably, in spores, GerRB-SGFP2 expressed under IPTG control was relatively uniformly distributed, and minimal if any full-length fusion protein was observed by Western blot analysis. These results suggest that when GerRB alone is overexpressed as a fusion with SGFP2: (1) the fusion protein is unstable, and (2) sporulating cells cannot localize high quantities of GerRB-SGFP2 in spores’ IM. Indeed, the fusion protein may form inclusion bodies when expressed in growing cells. Another possible concern is that use of the multicopy pDG148 leads to very high GerRB-SGFP2 expression at high IPTG levels, and perhaps expression with lower levels of IPTG might allow proper fusion protein location in spores [35]. However, the mechanism for integration of introduced genes into the *B. cereus* 14579 chromosomes is still to be determined. Additionally, suitable conditions for IPTG-inducible overexpression of spores’ GR in *B. cereus* are not entirely clear, and this is again a matter for further study.

## Figures and Tables

**Figure 1 microorganisms-10-01774-f001:**
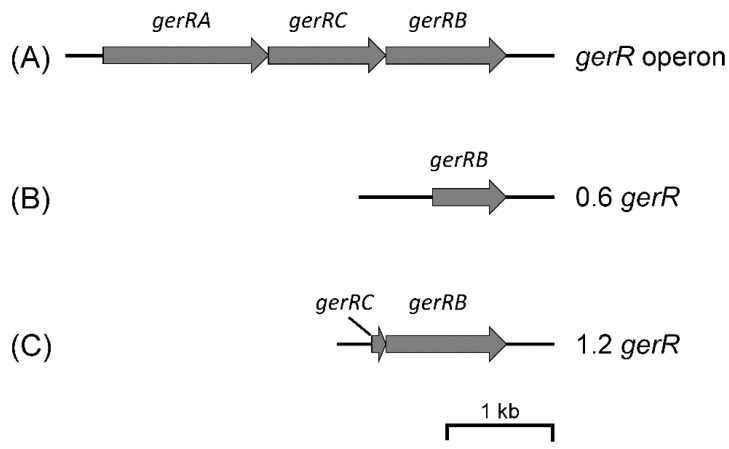
The map of the *gerR* operon and the locations of arms with *gerR* homology used in this work. (**A**) The *gerR* operon, consisting of *gerRA*, *gerRC,* and *gerRB*. (**B**) 0.6 *gerR*, a 678 bp fragment of the *gerRB* gene. (**C**) 1.2 *gerR*, a 1234 bp fragment of the *gerR* operon. The scale bar is 1 kb.

**Figure 2 microorganisms-10-01774-f002:**
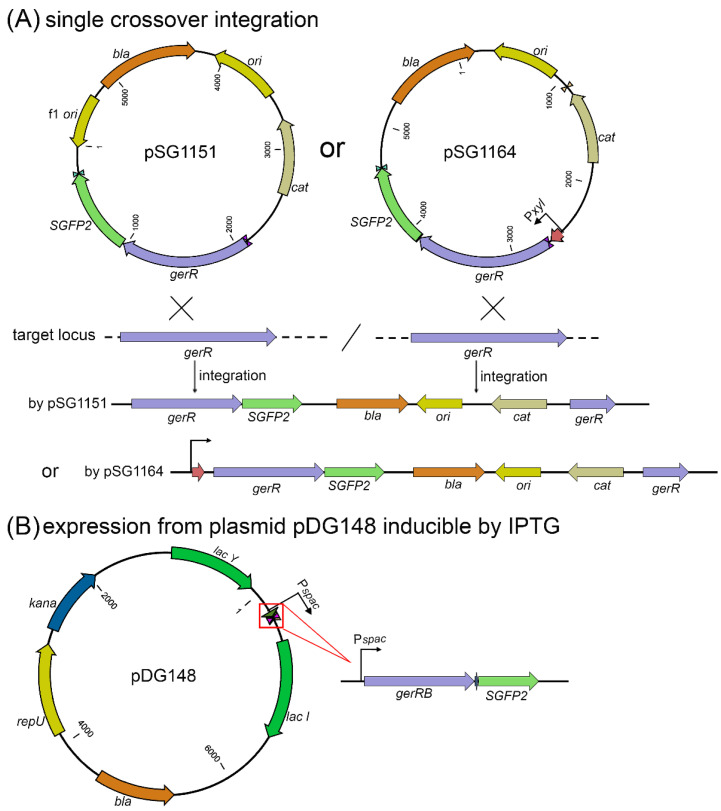
The workflow to construct integration and expression recombinant plasmids in this study. (**A**) The expected single crossovers into the *B. cereus* genome with integration vectors pSG1151 and pSG1164. Note that pSG1164 contains a xylose-inducible promoter. (**B**) The map of vector pDG148 with the *gerRB*-*SGFP2* fusion gene driven by the P*spac* promoter and inducible by IPTG.

**Figure 3 microorganisms-10-01774-f003:**
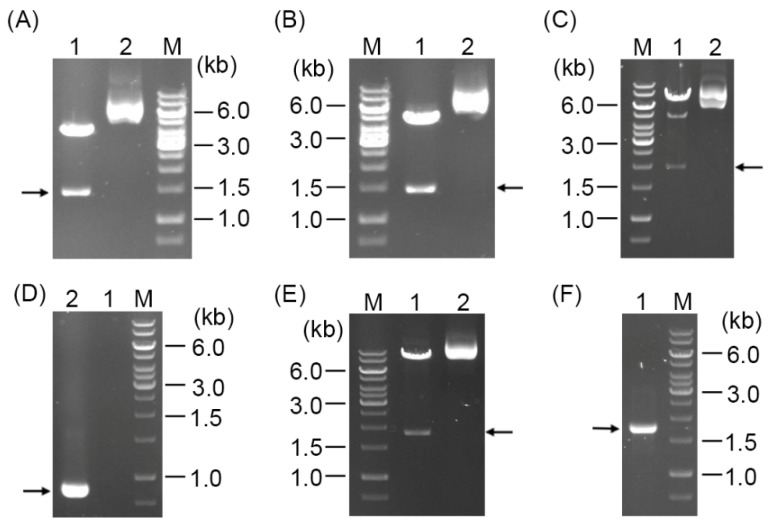
Confirmation of the structures of recombinant plasmids and the pDG148-containing strain of *B. cereus*. (**A**) Digestion of plasmid pSG1151-0.6 *gerR*-*SGFP2* with *Kpn* I and *Spe* I. Lane 1—the black arrow denotes the expected ~1.5 kb size of the 678 bp *gerRB* fused to *SGFP2*; lane 2—intact plasmid. (**B**) Digestion of plasmid pSG1164-0.6 *gerR*-*SGFP2* with *Kpn* I and *Spe* I. Lane 1—the black arrow denotes the expected ~1.5 kb size of the 678 bp *gerRB* fused to *SGFP2*; lane 2—intact plasmid. (**C**) Digestion of plasmid pSG1164-1.2 *gerR*-*SGFP2* with *Kpn* I and *Spe* I. Lane 1—the black arrow denotes the expected size of the 1234 bp *gerR* fused to *SGFP2* (~1.8kb); lane 2—intact plasmid. (**D**) Analysis of the integration of plasmid pSG1164-1.2 *gerR*-*SGFP2* in *B. cereus*. Lane 1—the expected 1.6 kb integrated fragment of 1.2 *gerR*-*SGFP2* was not present; lane 2—the black arrow indicates the expected size of the 870 bp *gerRB* fragment. (**E**) Digestion of plasmid pDG148-*gerRB*-*SGFP2* with *Sal* I and *Sph* I. Lane 1—the black arrow denotes the expected ~1.8 kb size of the 1110 bp *gerRB* fragment fused to *SGFP2*; lane 2—intact plasmid. (**F**) Colony PCR of *B. cereus* carrying plasmid pDG148-*gerRB*-*SGFP2*. Lane 1—*B. cereus* plasmid; the black arrow indicates the expected size of the 1.8 kb *gerRB*-*SGFP2* fusion. Lane M in all panels is the GeneRuler 1 kb DNA ladder.

**Figure 4 microorganisms-10-01774-f004:**
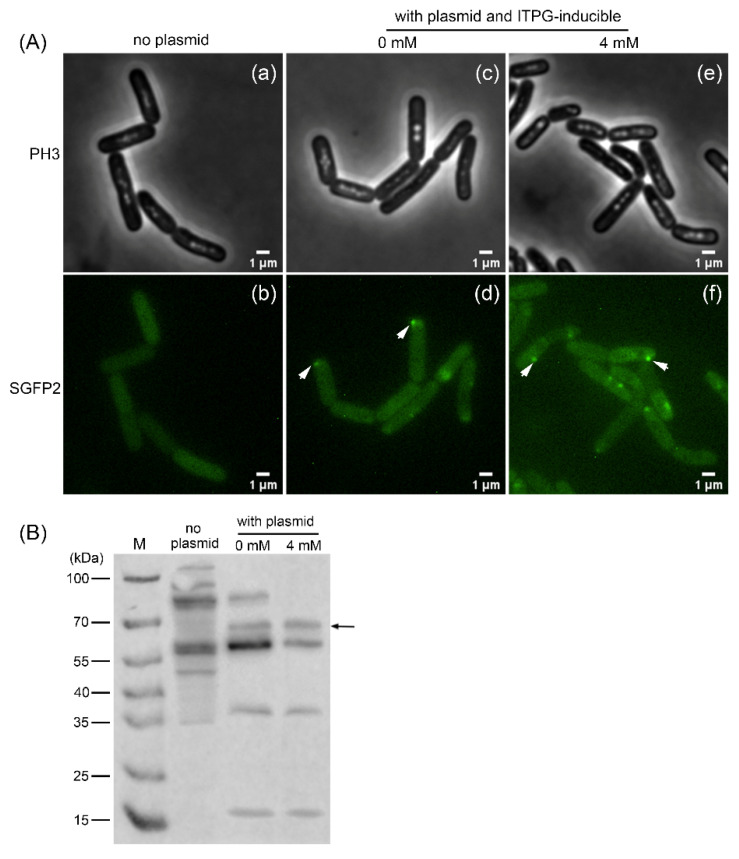
Visualization and detection of GerRB-SGFP2 in vegetative *B. cereus* cells containing plasmid pDG148-*gerRB*-*SGFP2*. (**A**) Phase contrast (PH3) microscopy (panels Aa–Ac) and (SGFP2) fluorescence microscopy (panels Ad–Af) of *B. cereus* vegetative cells with or without plasmid pDG148-*gerRB*-*SGFP2*. Panels Aa and Ab—cells without plasmid. Panels Ac and Ad—cells with plasmid but grown without IPTG (0 mM). Panels Ae and Af—cells containing plasmid grown with 4 mM IPTG. (**B**) Detection of GerRB-SGFP2 expression by Western blot analysis. Lane M, PageRuler™ Prestained Protein Ladder; no plasmid lane—protein of wild type cells; 0 mM lane—protein of cells containing plasmid and grown without IPTG; and 4 mM lane—protein of cells containing plasmid and grown with 4 mM IPTG. White arrows in A indicate foci of the GerRB-SGFP2 fusion protein. The black arrow in B denotes the expected size of the GerRB-SGFP2 protein at ~70 kDa.

**Figure 5 microorganisms-10-01774-f005:**
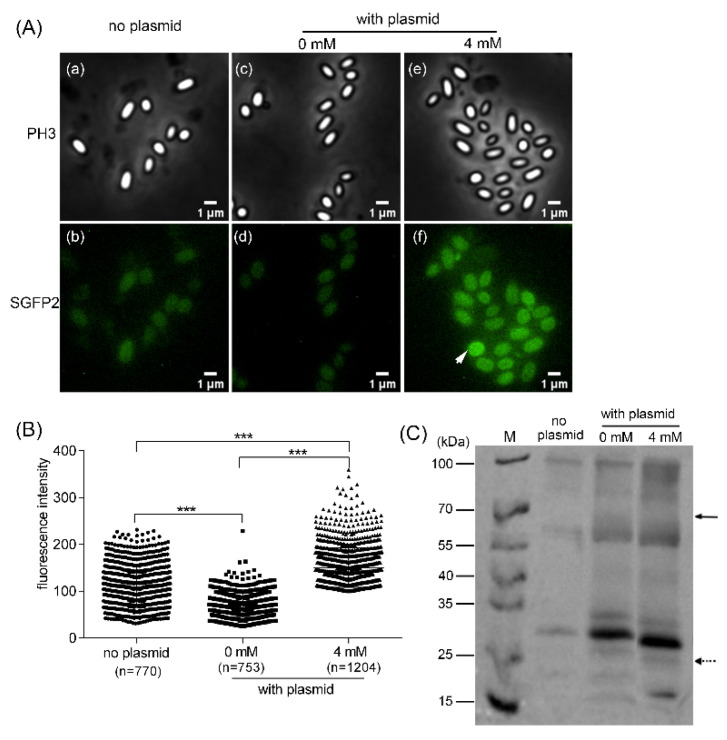
Imaging using phase contrast (PH3) or fluorescence (SGFP2) microscopy of GerRB-SGFP2 in *B. cereus* spores with or without plasmid pDG148-*gerRB*-*SGFP2* and prepared with or without 4 mM IPTG. (**A**) Visualization of dormant spores of *B. cereus*. Panels Aa and Ab—wild type *B. cereus* (no plasmid) spores. Panels Ac and Ad—dormant plasmid-containing spores prepared without IPTG. Panels Ae and Af—dormant spores containing plasmid and prepared with 4 mM IPTG. Panels Aa, Ac, and Ae—phase contrast images. Panels Ab, Ad, and Af—SGFP2 green fluorescence. (**B**) The fluorescence intensity of many hundreds of individual wild type and plasmid-carrying spores, the latter prepared with or without IPTG. (**C**) Western blot analysis of levels of GerRB-SGFP2 in spores. Lane M, PageRuler™ Pre-stained Protein Ladder; no plasmid lane—total protein of wild type spores; 0 mM lane—total protein of spores with plasmid and prepared without IPTG; 4 mM lane—total protein of spores with plasmid and prepared with 4 mM IPTG. The white arrow in panel A denotes a spore with high fluorescence. The black arrow in panel C denotes the expected location of the GerRB-SGFP2 fusion protein. The dotted arrow in panel C denotes the possible location of protein SGFP2. Statistical significance in panel B was calculated by one-way analysis of variance; *** *p* < 0.001.

**Table 1 microorganisms-10-01774-t001:** Strains and plasmids used or created in this study *.

Strains or Plasmids	Description	Sources
**Strains**		
*B. cereus* ATCC 14579	*B. cereus* wild type	Lab stock
*B. cereus* pDG148-GerRB-SGFP2	*B. cereus* containing plasmid pDG148-*gerRB*-*SGFP2* Kan^r^	This study
**Plasmids**		
pSGFP2-C1	Source of the *SGFP2* gene Kan^r^	[20]
pSG1151	Integration vector for *B. subtilis* Amp^r^ Chl^r^	[18]
pSG1151-0.6 *gerR*-*SGFP2*	678 base homology arm of *gerR* fused to *SGFP2* Amp^r^ Chl^r^	This study
pSG1151-1.2 *gerR*-*SGFP2*	1234 base homology arm of *gerR* fused to *SGFP2* Amp^r^ Chl^r^	This study
pSG1164	Integration vector with xylose inducible promoter in *B. subtilis* Amp^r^ Chl^r^	[18]
pSG1164-0.6 *gerR*-*SGFP2*	678 base homology arm of *gerR* fused to *SGFP2* Amp^r^ Chl^r^	This study
pSG1164-1.2 *gerR*-*SGFP2*	1234 base homology arm of *gerR* fused to *SGFP2* Amp^r^ Chl^r^	This study
pDG148	IPTG-inducible shuttle vector in *E. coli* and *B. subtilis* Amp^r^ Kan^r^	
pDG148-*gerRB-SGFP2*	*gerRB-SGFP2* fusion gene under control of the P*spac* promoter and inducible by IPTG Amp^r^ Kan^r^	This study

* The abbreviations used are: Kan^r^—resistant to kanamycin; and Amp^r^—resistant to ampicillin.

**Table 2 microorganisms-10-01774-t002:** Primers used in this study.

Name	Primers (5′-3′)
YW_1	GGGGTACCGGGATATGTTTTTGGGGT
YW_2	CTCGCCCTTGCTCACCATaccagcaccAGGTGTATCGGTTGAAGA *
YW_3	TCTTCAACCGATACACCTggtgctggtATGGTGAGCAAGGGCGAG *
YW_4	GGACTAGTTTACTTGTACAGCTCGTCCAT **
YW_5	GGGGTACCAGAAGCCATACGAGAAAA **
YW_6	CCCGTCGACGAGGTGAAATGAGCAATGAA **
YW_7	CCCGCATGCTTACTTGTACAGCTCGTCCAT **
YW_8	CAAAGACCCCAACGAGAA
YW_9	AAGTATGTCGAAAGGGGAA
YW_10	CACTATATATCCGTGTCGTT
YW_11	AAGGCGATTAAGTTGGGT
YW_12	CTACATCCAGAACAACCTCTGC
YW_13	TTCGGAAGGAAATGATGACCTC

* The flexible linker Gly-Ala-Gly coding sequence is written in lower-case letters. ** Restriction enzyme cleavage sites are underlined.

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
