# Peer review of "Genomic versus Plasmid-Borne Expression of Germinant Receptor Proteins in Bacillus cereus Strain 14579"

_microorganisms, 2022, doi:10.3390/microorganisms10091774_

Round 1
Reviewer 1 Report
Dear Authors,
all comments and suggestions is in appendix.
Best Regards

Reviewer 2 Report
The manuscript describe 3 plasmids with fragments of gerR operon used during B. cereus transformation. Although the work is well performed and easy to follow, additional experimental work is necessary.
1.- The occurrence of a low level of transformation (one to six colonies per microgram of DNA) in B cereus has been demonstrated. Therefore, is necessary additional effort to support the conclusions, such as additional B. cereus strains (wt, mutants or expressing comK, additional plasmids with different genes as controls, different electroporation conditions and DNA concentration. Also, use different transformation methods and mono-, di- or multimer forms of plasmid DNA.
Reviewer 3 Report
Please refer to the attached file for my comments.

Round 2
Reviewer 2 Report
Dear authors,
I agree with your answers to my comments.
I have only minor comments, in figure 3D, change the text for line 1 and 2, I think that the text for line 1 and 2 are upside down.
Author Response
"Please see the attachment"

Reviewer 3 Report
I appreciate the effort made by the authors to provide responses to the question raised by me in my previous review. The authors have provided adequate responses to most of the questions to my satisfaction and made the necessary amendments. The overall quality of the manuscript has been improved.
I disagree with authors at few points:
a) the author's mentioned in rebuttal that they have not been able to find a better antibiotic than chloramphenicol for such purpose. Why the authors not stick to the same antibiotic (either chloramphenicol or kanamycin) for plasmid and integrative plasmids, instead of using different ones for comparison.
b) Western blot could be standardized to eliminate the non-specific binding of antibody. More washing steps, shorter incubation, dilution etc could help.
c) The appropriate control for background fluoresence is the strain harboring empty vector. And for microscopy image number of cells in frame does matter, more the cells the image will look more bright and can be adjusted to look more bright. These results could be supported by doing GFP measurement using spectroscopy to eliminate any such doubts.
